# OpenReview forum: "UniFLoW: Universal Multi-Modal Federated LoRA Fine-Tuning Framework with Analytical Aggregation"
_ICLR.cc/2026/Conference — Submitted to ICLR 2026_

### Official Review · Reviewer_2BaQ · 2025-10-24

**Soundness:** 3
**Presentation:** 3
**Contribution:** 2
**Rating:** 4
**Confidence:** 4

**Summary:**

This work proposes UniFLoW (Universal Multi-modal Federated LoRA Fine-tuning Framework with Analytical Aggregation), a unified federated framework that leverages pre-trained large models and a multi-modal architecture. Moreover, it introduces Federated Aggregating Analytical Low-Rank Adaptation (FedA2-LoRA), which directly averages \\( A^t \\) to obtain \\( A^{t+1} \\), and then recovers the corresponding \\( B^{t+1} \\) matrices from the aggregated update \\( \Delta_W^* \\) using a closed-form solution of regularized least squares regression (Ridge Regression).

**Strengths:**

- The introduced FedA2-LoRA is both novel and interesting, effectively addressing aggregation errors in FL with LoRA fine-tuning.
- The paper is well-written and clearly articulated, making it easy to understand.

**Weaknesses:**

- This work exaggerates its contributions. The advantage of UniFLoW in addressing architectural incompatibility when dealing with multimodal data (Problem 2) stems from the characteristics of the modality-specific encoder (ImageBind [1]), which can handle various modalities, rather than from the contributions of this work.
- The proposed UniFLoW is based on specific encoders (ImageBind [1]) and LLMs (Vicuna-7B [2]). Can different encoders and LLMs be used?
- Why are the experimental results presented in Table 5 much worse than the results presented in Table 1 in FedSA-LoRA [3]?



[1] Rohit Girdhar, Alaaeldin El-Nouby, Zhuang Liu, Mannat Singh, Kalyan Vasudev Alwala, Armand Joulin, and Ishan Misra. Imagebind: One embedding space to bind them all. In Proceedings of the IEEE/CVF conference on computer vision and pattern recognition, pp. 15180–15190, 2023.

[2] Lianmin Zheng, Wei-Lin Chiang, Ying Sheng, Siyuan Zhuang, Zhanghao Wu, Yonghao Zhuang, Zi Lin, Zhuohan Li, Dacheng Li, Eric Xing, et al. Judging llm-as-a-judge with mt-bench and chatbot arena. Advances in neural information processing systems, 36:46595–46623, 2023.

[3] Pengxin Guo, Shuang Zeng, Yanran Wang, Huijie Fan, Feifei Wang, and Liangqiong Qu. Selective aggregation for low-rank adaptation in federated learning. arXiv preprint arXiv:2410.01463, 2024.

**Questions:**

- In the first stage, when the number of local iteration steps is less than τ, the model updates only the parameters of the corresponding encoder. When the steps exceed τ, the model updates only the parameters of the LLMs (Lines 242-245). What would be the effect of training the LLMs first and then training the encoder?
- Why is it better to train the LLM and the encoder in an II-stage approach rather than training both simultaneously? Is this related to the statement: "However, in FL, when client data exhibits certain biases, only specific types of multimodal data may be available. If
the encoder is not fine-tuned, this data can influence the fine-tuning of the base model, causing it to specialize for a specific modality and thus negatively impacting the model’s generalization." (Lines 238-241) Thus, what would be the effect of training the encoder for the first T communication rounds and then training the LLM for the following T rounds?
- What is the time complexity of solving Equation 11? Since it involves matrix inversion, how does the time complexity look?
- Line 373, "Please refer to the Appendix for a detailed evaluation." I did not find it in the Appendix.

---

> ### Author Response · Authors · 2025-11-21
> **Author Responses (1/3)**
>
> We sincerely thank you for the valuable comments and constructive feedback. Your insights have greatly helped us identify unclear explanations, missing ablation analyses, and opportunities to strengthen the theoretical justification. In the revised manuscript, we have carefully addressed and resolved all issues to the best of our ability. Below, we provide point-by-point responses to each of your comments.
>
> >W1: This work exaggerates its contributions. The advantage of UniFLoW in addressing architectural incompatibility when dealing with multimodal data (Problem 2) stems from the characteristics of the modality-specific encoder (ImageBind [1]), which can handle various modalities, rather than from the contributions of this work.
>
> Thank you for your insightful comment and for giving us the opportunity to clarify the contribution of UniFLoW. We agree that our framework builds upon a strong modality encoder, ImageBind [1], and we fully acknowledge its ability to process diverse input modalities. However, our contribution lies in proposing a *universal multimodal federated fine-tuning framework* that leverages existing modality encoders and pretrained models—an approach that, to the best of our knowledge, has not been explored in prior work. We hope UniFLoW can serve as a foundation for future FedMLLM fine-tuning paradigms.
>
> In addition, UniFLoW addresses several challenges that emerge after integrating such multimodal encoders into a federated setting.
> First, to mitigate performance degradation caused by modality imbalance during federated training, we introduce a  Ⅱstage training strategy that stabilizes optimization across heterogeneous clients.Second, to resolve the aggregation error inherent in standard LoRA under federated updates, we propose $FedA^2-LoRA$, which enables accurate parameter aggregation while significantly reducing communication overhead.These capabilities cannot be achieved by ImageBind alone, and together they form the core technical contribution of UniFLoW.
>
> >W2: The proposed UniFLoW is based on specific encoders (ImageBind [1]) and LLMs (Vicuna-7B [2]). Can different encoders and LLMs be used?
>
> Thank you for your question. The encoder and the pretrained large model in UniFLoW are, in fact, interchangeable. In our experiments, we further validate this by using UniBind[4] and LLaMA-1B[5] as the client models, and we report the corresponding results as follows,We have also added the table to **Appendix C.6**.
>
> | Method| Image Acc | Image P_BERT | Image R_BERT | Image F_BERT | Audio Acc| Audio P_BERT| Audio R_BERT| Audio F_BERT|
> |-|-|-|-|-|-|-|-|-|
> |–| 52.17 | 80.85 | 82.47| 81.61| 52.94| 81.92| 79.36 | 80.61|
> |LoRA | 63.15 | 83.8| **83.55**| 83.67| 61.90|  85.16  | 77.49| 81.14 |
> |FedA²-LoRA  | 65.00 | **88.41**| 81.06| **84.58** | 62.50| 84.29 | 82.57 | 83.41 |
> |FedA²-LoRA (Ⅱ stage) | **66.67** | 83.77 | 83.39 | 83.58| **63.63**| **86.01**| **84.01** | **85.00**|
>
> >W3: Why are the experimental results presented in Table 5 much worse than the results presented in Table 1 in FedSA-LoRA [3]?
>
> Thank you for your question. This part of our implementation follows the approach in [3], and we sincerely appreciate their contribution. Overall, most of our experimental settings and hyperparameters differ from theirs. The main differences are summarized as follows:
>
> | parameter| batch size| learning rate |
> |-|-|-|
> | FedSA-LoRA| 128 | 5E-3, 1E-2, 2E-2, 5E-2, 1E-1 |
> | FedA²-LoRA | 64| 0.1 |
>
> The main difference is that [3] states in the paper: “We report **the best result** from experiments run with learning rates η ∈ {5E-3, 1E-2, 2E-2, 5E-2, 1E-1}.”
> In contrast, we adopt a fixed learning rate of 0.1 and report the **final** performance rather than the best performance. Although we also observe peak accuracies exceeding 90+, such high values may not consistently appear under different random seeds. Therefore, we choose to report the final results.
>
> [1] Rohit Girdhar, Alaaeldin El-Nouby, Zhuang Liu, Mannat Singh, Kalyan Vasudev Alwala, Armand Joulin, and Ishan Misra. Imagebind: One embedding space to bind them all. In Proceedings of the IEEE/CVF conference on computer vision and pattern recognition,2023.
>
> [2] Lianmin Zheng, Wei-Lin Chiang, Ying Sheng, Siyuan Zhuang, Zhanghao Wu, Yonghao Zhuang, Zi Lin, Zhuohan Li, Dacheng Li, Eric Xing, et al. Judging llm-as-a-judge with mt-bench and chatbot arena. Advances in neural information processing systems, 2023.
>
> [3] Pengxin Guo, Shuang Zeng, Yanran Wang, Huijie Fan, Feifei Wang, and Liangqiong Qu. Selective aggregation for low-rank adaptation in federated learning. arXiv preprint arXiv:2410.01463, 2024.
>
> [4]Lyu Y, Zheng X, Zhou J, et al. Unibind: Llm-augmented unified and balanced representation space to bind them all[C]//Proceedings of the IEEE/CVF Conference on Computer Vision and Pattern Recognition. 2024.
>
> [5]Zhao J, Zhang Z, Chen B, et al. Galore: Memory-efficient llm training by gradient low-rank projection[J]. arXiv preprint arXiv:2403.03507, 2024.

---

> ### Author Response · Authors · 2025-11-21
> **Author Responses (2/3)**
>
> >Q1: In the first stage, when the number of local iteration steps is less than τ, the model updates only the parameters of the corresponding encoder. When the steps exceed τ, the model updates only the parameters of the LLMs (Lines 242-245). What would be the effect of training the LLMs first and then training the encoder?
>
> Thank you for this insightful question. Our intuition is that first calibrating the modality space in the first stage, and then allowing the LLM to learn semantics on top of the stabilized representations in the second stage, is a reasonable assumption. Reversing this order would break this dependency. We therefore provide additional experiments below to show that such a reversal brings only marginal benefits,and we have also added the table to **Appendix C.8**.
>
> | Method   | Image Acc | Image P_BERT | Image R_BERT | Image F_BERT | Audio Acc | Audio P_BERT | Audio R_BERT | Audio F_BERT |
> |-|-|--|-|--|--|-|--|-|
> | –       | 61.23     | 78.59        | 81.74        | 80.09        | 50.19     | 78.96        | 81.71        | 80.02        |
> | LoRA    | 64.89     | 86.13        | 82.22        | 84.11        | 58.33     | 82.42        | **87.75**        | 84.97        |
> | FedA²-LoRA   | 75.00     | 84.51        | 84.34        | 84.43        | 60.17     | 84.97        | 85.83        | 85.39        |
> | FedA²-LoRA (Reverse)    | 69.63 | **86.71**     | 81.09    |   83.81    | 60.61    | **87.33**  | 82.94    | 85.07 |
> | FedA²-LoRA  (Ⅱ stage)    | **76.19**| 83.44| **88.08**| **85.69**| **60.90**| 87.15| 85.49| **86.31**    |
>
> >Q2: Why is it better to train the LLM and the encoder in an II-stage approach rather than training both simultaneously? Is this related to the statement: "However, in FL, when client data exhibits certain biases, only specific types of multimodal data may be available. If the encoder is not fine-tuned, this data can influence the fine-tuning of the base model, causing it to specialize for a specific modality and thus negatively impacting the model’s generalization." (Lines 238-241) Thus, what would be the effect of training the encoder for the first T communication rounds and then training the LLM for the following T rounds?
>
> Thank you for your valuable question. We would like to clarify that our two-stage design is indeed motivated by the phenomenon described in Lines 238–241, and we further explain why two-stage training is superior to joint synchronous training.
>
> In heterogeneous multimodal federated learning, different clients hold different modalities (image, audio, text). If we train the encoder and the LLM simultaneously, each modality produces inconsistent hidden representations, causing the client-side LLM updates to conflict with each other. As a result, the LLM tends to overfit modality-specific noise, which significantly harms both convergence and generalization.
>
> Regarding your question about *“what would be the effect of training the encoder for the first T communication rounds and then training the LLM for the following T rounds”*, we conducted experiments to evaluate this setting. The results are shown below and we have also added the table to **Appendix C.9**.CR indicates the total number of communication rounds.
>
> | Method   | Image Acc | Image P_BERT | Image R_BERT | Image F_BERT | Audio Acc | Audio P_BERT | Audio R_BERT | Audio F_BERT |
> |-|-|--|-|--|--|-|--|-|
> | –       | 61.23     | 78.59        | 81.74        | 80.09        | 50.19     | 78.96 | 81.71   | 80.02 |
> | LoRA   | 64.89     | **86.13**        | 82.22        | 84.11        | 58.33     | 82.42  | **87.75**        | 84.97        |
> | FedA²-LoRA   | 75.00     | 84.51        | 84.34        | 84.43        | 60.17     | 84.97| 85.83  | 85.39        |
> | FedA²-LoRA (T=0.5CR)   | 66.15 |84.03|85.78|84.90 | 60.35  | **88.98**  | 82.72 | 85.73  |
> | FedA²-LoRA  (Ⅱ stage)  | **76.19**| 83.44| **88.08**| **85.69**| **60.90**| 87.15| 85.49| **86.31**    |

---

> ### Author Response · Authors · 2025-11-21
> **Author Responses (3/3)**
>
> >Q3: What is the time complexity of solving Equation 11? Since it involves matrix inversion, how does the time complexity look?
>
> Thank you for this helpful question. Equation (11) corresponds to solving a small ridge-regression problem in closed form. Concretely, if the LoRA matrix $A \in \mathbb{R}^{r \times d}$, then the server needs to invert the matrix $A A^{\top} \in \mathbb{R}^{r \times r}$.This is also one of the reasons why we choose to aggregate $A$ and solve for $B$. If we were to compute the inverse involving $BB^{\top}$, the computational complexity would increase to $O(d^3)$.
>
>
> The **matrix inversion** therefore has time complexity $\mathcal{O}(r^{3})$,where $r$ is the LoRA rank (typically a small constant such as 4, 8, or 16 in our experiments).
> The surrounding matrix multiplications (e.g., computing $U A^{\top}$ and $A A^{\top}$) contribute terms on the order of $\mathcal{O}(d r^{2}$), which are also low-order compared to local forward–backward passes on the full model.
>
> Since $d$ typically ranges from 1024 to 4096 and $r \ll d$, the dominant term in the overall complexity is the computation of $\Delta_W^*$.
> We have added this complexity analysis to the revised manuscript (**Appendix C.1**).We also provide, below, a comparison of computational complexity, training parameters, and communication parameters.
>
> | Method | # Trainable Params | # Per-round Communicated Params | Computational Complexity|
> |-|-|-|-|
> |LoRA |1.83M| 0.78M|O(Krd)|
> |FFA-LoRA|1.44M| 0.39M |O(Krd) |
> |FedDPA-LoRA  | 2.62M  | 0.78M | O(Krd) |
> |FedSA-LoRA| 1.83M | 0.39M | O(Krd) |
> |FedEx-LoRA|1.83M | 25.74M | O(Krd²) |
> |$FedA²-LoRA$|1.83M | 0.78M|O(Krd²)|
>
> >Q4: Line 373, "Please refer to the Appendix for a detailed evaluation." I did not find it in the Appendix.
>
> We apologize for the confusion caused by our previous wording. Our intention was to direct the reviewer to the dataset descriptions and the detailed explanation of the evaluation metrics. We have now revised the main text to: *“The detailed descriptions of the datasets and evaluation metrics are provided in Appendix G.”*
>
> We sincerely appreciate your insightful feedback and have addressed your concerns to the best of our ability. Should you have any further questions or recommendations, we would be glad to engage in further discussion.😁

---

> > ### Author Response · Authors · 2025-11-25
> > **Looking forward to your feedback**
> >
> > Dear Reviewer 2BaQ,
> >
> > We are eager to ensure that we have adequately addressed your concerns and are prepared to offer further clarifications or address any additional questions you may have.
> >
> > Should you find that our revisions have satisfactorily addressed your main concerns, we would be most grateful if you would **reconsider the evaluation of our paper to enhance its standing**.
> >
> > We would like to express our heartfelt gratitude for the time and effort you have dedicated to reviewing our work. It has been a pleasure to engage with you throughout this process.😁
> >
> > Best regards,
> >
> > The authors

---

### Official Review · Reviewer_ZbHw · 2025-10-30

**Soundness:** 2
**Presentation:** 2
**Contribution:** 3
**Rating:** 4
**Confidence:** 5

**Summary:**

To efficiently leverage distributed multimodal data under heterogeneous multimodal settings, we propose FedA²-LoRA within the FL and MLLMs framework. The method adopts a two-stage training strategy—first fine-tuning the modality-specific encoder’s LoRA, followed by the LLM’s LoRA and introduces Tikhonov regularization on the LoRA A matrix to approximate the B matrix, thereby improving aggregation consistency. Experimental results demonstrate the effectiveness of the proposed method.

**Strengths:**

The method takes into account the issues of modality heterogeneity and the aggregation bias between the LoRA A and B matrices, making its research motivation reasonably meaningful.

**Weaknesses:**

1. The approach of approximating (B) from (U) and (A) (Equations 9–11) is not very reasonable. If each client needs to upload both (B) and (A) to the server to compute (U), then it would be more straightforward to directly multiply (B) and (A) on the server and aggregate the results, which would inherently avoid the aggregation inconsistency. Moreover, uploading both (B) and (A) does not actually reduce the communication cost.

2. The use of Tikhonov Regularization to approximate matrix (B) lacks theoretical justification, making the approach less convincing.

3. The results in Tables 2–4 seem to show only that the two-stage training strategy performs better than the single-stage approach that trains the modality encoder and LLM LoRA simultaneously. While this two-stage strategy could be an effective training method, it may not be sufficient to constitute a complete innovation.

4. There are no additional ablation studies to verify the effectiveness of using Tikhonov Regularization for approximating matrix (B).

5. Figures 1 and 2 are not clearly presented—particularly Figure 2, which is overly complicated and fails to highlight the key points. In addition, the overall writing quality of the paper still needs improvement.

**Questions:**

The main issues are that the approach for approximating (B) appears unreasonable and does not actually reduce communication costs. The experimental results mainly highlight the effect of the two-stage training strategy; however, this strategy alone is insufficient to constitute a complete innovation, and the method for approximating (B) also lacks theoretical justification.

---

> ### Author Response · Authors · 2025-11-21
> **Author Responses (1/2)**
>
> We sincerely thank you for the thoughtful and constructive feedback. Your comments have helped us identify unclear explanations, strengthen the theoretical justification, and improve the overall presentation. We have carefully addressed all concerns in the revised version.
> >W1&Q: The approach of approximating (B) from (U) and (A) (Equations 9–11) is not very reasonable. If each client needs to upload both (B) and (A) to the server to compute (U), then it would be more straightforward to directly multiply (B) and (A) on the server and aggregate the results, which would inherently avoid the aggregation inconsistency. Moreover, uploading both (B) and (A) does not actually reduce the communication cost.
>
> Thank you for pointing out the issue. Due to our writing inadequacy, it led to a misunderstanding. You are correct that directly solving $U=\sum B _ iA _ i$ on the server can eliminate the aggregation error. However, when transmitting back to the client, two issues arise. First, if we don't decompose into $B$ and $A$, subsequent updates become impossible. Second, for a large model like 7B, where $U \in \mathbb{R}^{d \times d}$,$B \in \mathbb{R}^{d \times r} $, and $A \in \mathbb{R}^{r \times d} $ with $d = 4096 $ and $ r = 4, 8, 16 $, the communication required to transmit $U$ is 512–128 times larger than the communication required for $B$ and $A$. Therefore, we propose $FedA^2-LoRA$ to reduce communication costs while ensuring accuracy.
>
> Furthermore, regarding your comment that our method does not reduce communication costs, we apologize for the confusion caused by the description in the previous version. We have revised this in the latest version. To clarify, our method reduces communication costs in comparison to the FedEx-LoRA method, which also recovers $U$. Compared to FedEx-LoRA, our approach saves 128 times the communication volume.
>
> >W2&Q: The use of Tikhonov Regularization to approximate matrix (B) lacks theoretical justification, making the approach less convincing.
>
> Thank you for your valuable feedback regarding Tikhonov Regularization to approximate matrix (B). We have supplemented the content on the optimization objective and the proof of optimality in both the main text and **Appendix B**. Here is a brief overview:
>
> On the server side, our optimization objective is formulated to minimize the aggregation error. Specifically, we seek $B$ and $A$ that satisfy:
>
> $\min _ B ||  B  A - U || _ F ^ 2$
>
> Because $AA^{T}$ may be ill‑conditioned or non‑invertible, we apply Tikhonov regularization to ensure stability, resulting in the modified objective:
>
> $\min_B ||  BA - U ||_F^2 + \lambda ||  B ||_F^2$
>
> By computing the derivative with respect to $B$ and setting it to zero, we obtain the closed‑form solutions:
>
> $B =U A^T [A A^T + \lambda I]^{-1}, \text{if } (A A^T)^{-1}  \text{does not exist}$
>
> $B =U A^T [A A^T ]^{-1}, \text{if } (A A^T)^{-1}  \text{exists}$
>
> These derivations and related discussions are now explicitly provided in the revised paper.
> >W3&Q: The results in Tables 2–4 seem to show only that the two-stage training strategy performs better than the single-stage approach that trains the modality encoder and LLM LoRA simultaneously. While this two-stage strategy could be an effective training method, it may not be sufficient to constitute a complete innovation.
>
> Thank you for your question. The main contribution of our work is the proposal of a federated, general-purpose, multimodal large model fine-tuning framework, UniFLow. This framework primarily relies on existing modality encoders and pre-trained large models, along with our proposed $FedA^2LoRA$ and a two-stage training strategy, to leverage client-side private data and enhance global model performance. Therefore, our contribution extends beyond just the two-stage training approach. Below, we present a comparison between $FedA^2-LoRA$ and other methods:
>
> | Method               | Image Acc | Image P_BERT | Image R_BERT | Image F_BERT | Audio Acc | Audio P_BERT | Audio R_BERT | Audio F_BERT |
> |-|-|--|-|--|--|-|--|-|
> | –       | 61.23     | 78.59        | 81.74        | 80.09        | 50.19     | 78.96        | 81.71        | 80.02        |
> | LoRA                 | 64.89     | **86.13** | 82.22        | 84.11        | 58.33     | 82.42        | **87.75**        | 84.97        |
> | FFA-LoRA             | 63.15     | 85.18        | 79.27        | 82.08        | 57.87     | **91.20**        | 77.04        | 83.26        |
> | FedSA-LoRA           | 66.29     | 84.22        | 83.15        | 84.22        | 58.33     | 82.32        | 78.93        | 83.67        |
> | FedEx-LoRA           | 66.67     | 81.21        | **85.81**        | **83.45**        | 58.33     | 83.92        | 83.82        | 83.79        |
> | FedA²-LoRA           | **75.00**     | 84.51        | 84.34        | 84.43        | **60.17**     | 84.97        | 85.83        | **85.39**        |

---

> > ### Author Response · Authors · 2025-11-21
> > **Author Responses (2/2)**
> >
> > >W4&Q: There are no additional ablation studies to verify the effectiveness of using Tikhonov Regularization for approximating matrix (B).
> >
> > Thank you for pointing out this issue. Here, we provide the additional ablation study on using Tikhonov Regularization for approximating the matrix (B), as shown below.
> >
> > | Method | Image Acc | Image P_BERT | Image R_BERT | Image F_BERT | Audio Acc | Audio P_BERT | Audio R_BERT | Audio F_BERT |
> > |----|---:|------:|-----:|-----:|------:|----:|-----:|---:|
> > | LoRA   | 64.89     | **86.13**        | 82.22        | 84.11        | 58.33     | 82.42        | **87.75**   | 84.97        |
> > | **FedA²-LoRA (λ = 0, w/o Tik.)**| 65.09    | 83.09 | 78.93        |80.96       | **62.64**    | 86.53         | 82.16     | 84.29       |
> > | **FedA²-LoRA (Tikhonov, λ=0.1)**| **76.19**    | 83.44        | **88.08**        | **85.69**        | 60.90     | **87.15**        | 85.49        | **86.31**   |
> >
> > Without regularization, the solved matrix may become ill-conditioned, which can lead to numerical overflow during multiple communication rounds or a drop in model performance.
> > At the same time, **Figure 3(a)** in the main text includes an ablation study on the hyperparameter $\lambda$. Below, we present the corresponding results on SST2, as follows.
> >
> > | Dataset | λ=0.1  | λ=0.5 | λ=1 | λ=1.5 | λ=2  |
> > |-|-|-|--|---|-|
> > | SST2   | 53.79  | 66.52  | 53.62 | 57.66 | 53.58  |
> >
> > >W5&Q: Figures 1 and 2 are not clearly presented—particularly Figure 2, which is overly complicated and fails to highlight the key points. In addition, the overall writing quality of the paper still needs improvement.
> >
> > Thank you for your valuable suggestion. In the revised version, we have simplified and redrawn Figure 2 by removing non-essential elements and enlarging the key components to more clearly highlight the main idea of our framework. The original version of Figure 2 has been moved to the appendix to help interested readers understand the detailed workflow of the model. In addition, we have carefully polished the writing throughout the paper, improved the logical flow, reduced redundancy, and standardized the terminology to enhance the overall readability.
> >
> > We sincerely appreciate your insightful feedback and have addressed your concerns to the best of our ability. Should you have any further questions or recommendations, we would be glad to engage in further discussion.😁

---

> ### Author Response · Authors · 2025-11-25
> **Looking forward to your feedback**
>
> Dear Reviewer ZbHw,
>
> We are eager to ensure that we have adequately addressed your concerns and are prepared to offer further clarifications or address any additional questions you may have.
>
> Should you find that our revisions have satisfactorily addressed your main concerns, we would be most grateful if you would **reconsider the evaluation of our paper to enhance its standing**.
>
> We would like to express our heartfelt gratitude for the time and effort you have dedicated to reviewing our work. It has been a pleasure to engage with you throughout this process.😁
>
> Best regards,
>
> The authors

---

### Official Review · Reviewer_TTmT · 2025-10-31

**Soundness:** 3
**Presentation:** 3
**Contribution:** 3
**Rating:** 4
**Confidence:** 5

**Summary:**

This paper proposes UniFLoW, a universal multi-modal federated LoRA fine-tuning framework targeting the challenges of applying Federated Learning (FL) to Multi-modal Large Language Models (MLLMs), namely client-side modality heterogeneity, high communication costs, and LoRA aggregation inconsistency. The central contribution is FedA²-LoRA, which aggregates client-side LoRA parameters by analytically reconstructing the  matrix via a closed-form ridge-regression solution. The framework adopts a two-stage training strategy that fine-tunes LoRA modules in both the modality encoders (ImageBind) and the base LLM (Vicuna-7B).Experiments on multi-modal QA tasks indicate its effectiveness.

**Strengths:**

1.FedA²-LoRA introduces an efficient analytical approach to address federated LoRA aggregation inconsistency. By directly aggregating the matrices and analytically recovering , it offers a communication-efficient alternative.
2.The work is significant in applying FL to MLLMs with heterogeneous modality data.
3.UniFLoW combines a general-purpose modality encoder (ImageBind) with an LLM and employs LoRA in key modules to cope with modality heterogeneity; the design rationale is clear and sensible.

**Weaknesses:**

1.FedA²-LoRA assumes “ is more global and is more local,” hence averaging and reconstructing from and . This rests on heuristic motivation; the paper should specify theoretical conditions under which the “global” nature of  holds, and whether it remains valid under non-IID settings.
2.The closed-form solution in Equation (11) essentially solves the ridge-regression problem , yet the paper does not explicitly present the objective nor provide a proof of optimality.
3.Although FedA²-LoRA is said not to increase communication costs, the experiments do not report measured communication budgets or parameter payloads.
4.The paper references FedEx-LoRA and related work but does not provide head-to-head comparisons under matched communication budgets and client participation. Current conclusions rely mainly on comparisons with FedSA and FFA and are therefore less convincing.
5.While the paper emphasises heterogeneous client resources, it does not propose explicit heterogeneity-aware mechanisms (e.g., variable-rank or variable-layer LoRA) that reflect realistic constraints.

**Questions:**

1.Mixed use of “Ⅱ/II stage”; please standardise.
2.Instances include “does not exists” in Equation (11) and “AAk” around line 300. There are occasional logical jumps and paragraph repetitions that reduce readability.
3.Please verify consistency of symbols and variable definitions throughout.

---

> ### Author Response · Authors · 2025-11-21
> **Author Responses (1/2)**
>
> Thank you for your detailed review and valuable comments on our paper. We have carefully read your feedback and have revised and supplemented it based on your suggestions. The following are our responses to your questions and suggestions:
> >W1: FedA²-LoRA assumes “ is more global and is more local,” hence averaging and reconstructing from and . This rests on heuristic motivation; the paper should specify theoretical conditions under which the “global” nature of holds, and whether it remains valid under non-IID settings.
>
> Thank you for your valuable feedback. We have already included the solution process for obtaining the optimal solution in the **Appendix A**. In this work, we assume that updates to A  and B are independent of each other. The optimal solutions for $A ^ *$ and $B ^ *$ are derived as shown below.
>
> $A^*= \mathbb{B}^{\dagger}\Delta_W$
>
> $B^*= \Delta_W\mathbb{E}[X_{n_k}X_{n_k}^T]\mathbb{A}^T (\mathbb{A}\mathbb{E}[X_{n_k}X_{n_k}^T]\mathbb{A}^T)^{-1}$
>
> Where $\mathbb{A}$ and $\mathbb{B}$ are Gaussian random initialization matrices.
> Additionally, under the experimental setting with non-IID (dir=0.5), we provide a comparison of the similarity between LoRA-A and LoRA-B at each layer for two clients. As shown in **Appendix Figure 4**, we sorted the similarity values and found that the similarity of  LoRA-A is significantly higher than that of LoRA-B. This indicates that A captures more general information, while B exhibits greater heterogeneity, suggesting it is more influenced by client-specific data. So we conclude that B captures more personalized information. And we show the similarity of the values ​​of each layer of LoRA below.
>
> | Layer Name | LoRA Parameter Name | Cosine Similarity |
> |:---|:---|:---|
> |15 | value.lora\_A | 0.212694 |
> |20 | value.lora\_A | 0.137847 |
> |12 | value.lora\_A | 0.117822 |
> |13 | value.lora\_A | 0.094493 |
> | 22 | value.lora\_A | 0.085506 |
> |18 | value.lora\_A | 0.083193 |
> |14 | value.lora\_A | 0.073243 |
> |19 | value.lora\_A | 0.062988 |
> |3 | value.lora\_A | 0.061706 |
> |21 | value.lora\_A | 0.048951 |
> |7 | value.lora\_A | 0.04726 |
> |1 | value.lora\_A | 0.038443 |
> |2 | value.lora\_A | 0.038439 |
> |16 | value.lora\_A | 0.035118 |
> |9 | value.lora\_A | 0.030625 |
> |4 | value.lora\_A | 0.029767 |
> |10 | value.lora\_A | 0.025706 |
> |10 | value.lora\_B | 0.025444 |
> |17 | value.lora\_B | 0.022276 |
> |23 | value.lora\_A | 0.019876 |
> |5 | value.lora\_A | 0.016306 |
> |14 | value.lora\_B | 0.015442 |
> |3 | value.lora\_B | 0.014453 |
> |11 | value.lora\_A | 0.01341 |
> |20 | value.lora\_B | 0.013018 |
> |0 | value.lora\_B | 0.010565 |
> |0 | value.lora\_A | 0.00969 |
> |6 | value.lora\_B | 0.007943 |
> |8 | value.lora\_A | 0.00707 |
> |1| value.lora\_B | 0.006977 |
> |22 | value.lora\_B | 0.006539 |
> |23 | value.lora\_B | 0.004681 |
> |11 | value.lora\_B | 0.004258 |
> |4 | value.lora\_B | 0.001911 |
> |18 | value.lora\_B | 0.001812 |
> |8 | value.lora\_B | 0.000097 |
> |5 | value.lora\_B | -0.001479 |
> |7 | value.lora\_B | -0.001675 |
> |19 | value.lora\_B | -0.002555 |
> |16 | value.lora\_B | -0.002883 |
> |21 | value.lora\_B | -0.003744 |
> |13 | value.lora\_B | -0.003967 |
> |17 | value.lora\_A | -0.005535 |
> |2 | value.lora\_B | -0.008209 |
> |15| value.lora\_B | -0.009123 |
> |9 | value.lora\_A | -0.011076 |
> |12 | value.lora\_B | -0.012479 |
>
> >W2: The closed-form solution in Equation (11) essentially solves the ridge-regression problem , yet the paper does not explicitly present the objective nor provide a proof of optimality.
>
> Thank you for your valuable feedback regarding the closed-form solution. **We have supplemented the content on the optimization objective and the proof of optimality in both the main text and Appendix B**. Here, I will provide a brief overview.
>
> On the server side, our optimization objective aims to minimize the aggregation error. Therefore, the desired $B$ and $A$ are as follows:
>
> $\min _ B ||  B  A - U || _ F ^ 2$
>
> Considering the instability of the pseudo-inverse of $A$, we applied regularization during the solution process for $B$. In this case, we modify the optimization objective to:
>
> $\min_B ||  BA - U ||_F^2 + \lambda ||  B ||_F^2$
>
> By taking the derivatives of the above optimization functions in turn and setting the derivatives equal to zero, we obtain the following results:
>
> $B =U A^T [A A^T + \lambda I]^{-1}, \text{if } (A A^T)^{-1}  \text{does not exist}$
>
> $B =U A^T [A A^T ]^{-1}, \text{if } (A A^T)^{-1}  \text{exists}$

---

> > ### Author Response · Authors · 2025-11-21
> > **Author Responses (2/2)**
> >
> > >W3: Although FedA²-LoRA is said not to increase communication costs, the experiments do not report measured communication budgets or parameter payloads.
> >
> > Thank you for your valuable feedback. We have added the training parameters, communication parameters, and the calculation of server complexity in **Appendix C.1**. Here, we provide you with a copy.
> >
> > | Method | # Trainable Params | # Per-round Communicated Params | Computational Complexity|
> > |-|-|-|-|
> > |LoRA |1.83M| 0.78M|O(2Krd)|
> > |FFA-LoRA|1.44M| 0.39M |O(Krd) |
> > |FedDPA-LoRA  | 2.62M  | 0.78M | O(Krd) |
> > |FedSA-LoRA| 1.83M | 0.39M | O(Krd) |
> > |FedEx-LoRA|1.83M | 25.74M | O(Krd²) |
> > |$FedA²-LoRA$|1.83M | 0.78M|O(Krd²)|
> >
> > >W4: The paper references FedEx-LoRA and related work but does not provide head-to-head comparisons under matched communication budgets and client participation. Current conclusions rely mainly on comparisons with FedSA and FFA and are therefore less convincing.
> >
> > Thank you for your valuable feedback. We have added FedEx-LoRA in Tables 2-4. Here, we only show the case where both the audio client and the image client are involved.
> >
> > | Method| Image Acc | Image P_BERT | Image R_BERT | Image F_BERT | Audio Acc | Audio P_BERT | Audio R_BERT | Audio F_BERT |
> > |-|-|-|-|-|-|-|-|-|
> > | FedEx‑LoRA | 66.67 | 81.21 | 85.81 | 83.45 | 58.33 | 83.92 | 83.82 | 83.79 |
> > | FedEx‑LoRA (II stage)| 69.40 | 81.89 | **88.06** | 84.86 | 59.76 | 82.78 | **86.50** | 84.60 |
> > | $FedA²‑LoRA$ | 75.00 | **84.51** |84.34| 84.43 | 60.17 | 84.97 | 85.83 | 85.39 |
> > | $FedA²‑LoRA $(II stage)| **76.19** | 83.44 | 88.08 | **85.69** | **60.90** | **87.15** | 85.49 | **86.31** |
> >
> > Above, we have shown the results with the same number of communication rounds, i.e., the same computational cost. We also find your suggestion reasonable. Below, we present a comparison of the global model performance under the same communication cost.We have also added the table to **Appendix C.3**.
> >
> > | Method|Rounds| Image Acc | Image P_BERT | Image R_BERT | Image F_BERT | Audio Acc | Audio P_BERT | Audio R_BERT | Audio F_BERT |
> > |-|-|-|-|-|-|-|-|-|-|
> > | LoRA | 33 |75.00 |80.14|86.12|83.03| 72.22 |79.88 |**87.46**| 83.46 |
> > | FFA-LoRA | 66 |**78.57**|86.74| **86.85**| **86.63**| 76.92 |85.18 |86.54| 85.72 |
> > | FedSA-LoRA|66 |72.72 |**86.88**| 84.35| 85.48| 78.26| **86.33**| 84.43 |85.37 |
> > | FedEx-LoRA |1| 61.53 | 79.78 | 80.91 | 80.31 | 55.56 |79.32 | 84.18 | 81.68 |
> > | $FedA²‑LoRA$|33 | 76.92 | 85.58 | 86.16 | 85.93 | **78.84** | 84.94 | 86.97 | **85.94** |
> >
> > >W5: While the paper emphasises heterogeneous client resources, it does not propose explicit heterogeneity-aware mechanisms (e.g., variable-rank or variable-layer LoRA) that reflect realistic constraints.
> >
> > Thank you for the reviewer's suggestion. Our UniFLoW focuses on handling heterogeneous modality clients. We believe that heterogeneity-aware mechanisms reduce computational costs by using different ranks for data from different clients, addressing the issue of heterogeneous data sizes. This is a real-world problem, and to demonstrate our method's adaptability to such scenarios, we have supplemented the experiments. Specifically, we set the ranks (r = 4) and (r = 8) for the encoders of five clients, and the results are as follows:
> >
> > | Method | Image Acc | Image P_BERT | Image R_BERT | Image F_BERT | Audio Acc | Audio P_BERT | Audio R_BERT | Audio F_BERT |
> > |-|-|-|-|-|-|-|-|-|
> > | - - |61.23| 78.59| 81.74| 80.09 |50.19 |78.96| 81.71 |80.02 |
> > |LoRA(zero-padding) |61.64|84.44| 80.38| 82.34| 56.27 |84.18 |**80.37**| 82.22 |
> > | $FedA²‑LoRA$(zero-padding)| **61.92** | **86.87** | **82.21** | **84.46** | **57.93** | **86.94** | 80.26 | **83.46** |
> >
> > We have also added the table to **Appendix C.4**.
> > >Q1: Mixed use of “Ⅱ/II stage”; please standardise.
> >
> > Thank you for pointing out the inconsistency in the usage of "Ⅱ/II stage" in our manuscript. We have conducted a thorough review and made the necessary corrections.
> > >Q2: Instances include “does not exists” in Equation (11) and “AAk” around line 300. There are occasional logical jumps and paragraph repetitions that reduce readability.
> >
> > Thank you for your careful review and for pointing out the errors. In the new version, we have corrected "exists" to "exist" and changed "$AA_k$" to "$A_k$". Additionally, to improve readability for the readers, we have made substantial revisions and reductions to the paper.
> >
> > >Q3: Please verify consistency of symbols and variable definitions throughout.
> >
> > Thank you for your suggestion regarding symbol and variable consistency.We have now conducted a thorough review of the entire manuscript to ensure that The same symbol is never used to represent two different quantities.For example, let $\mathcal{X}^{m} _ {n _ k}$ denote the feature representation and $X_{n_k}$ denote the prompt.
> >
> > We sincerely appreciate your insightful feedback and have addressed your concerns to the best of our ability. Should you have any further questions or recommendations, we would be glad to engage in further discussion.😁

---

> ### Author Response · Authors · 2025-11-25
> **Looking forward to your feedback**
>
> Dear Reviewer TTmT,
>
> We are eager to ensure that we have adequately addressed your concerns and are prepared to offer further clarifications or address any additional questions you may have.
>
> Should you find that our revisions have satisfactorily addressed your main concerns, we would be most grateful if you would **reconsider the evaluation of our paper to enhance its standing**.
>
> We would like to express our heartfelt gratitude for the time and effort you have dedicated to reviewing our work. It has been a pleasure to engage with you throughout this process.😁
>
> Best regards,
>
> The authors

---

### Official Review · Reviewer_oyVC · 2025-11-01

**Soundness:** 2
**Presentation:** 2
**Contribution:** 2
**Rating:** 4
**Confidence:** 3

**Summary:**

This paper addresses the challenge of fine-tuning Multimodal Large Language Models in federated learning settings. The authors explain that applying traditional FL to MLLMs can be very expensive and methods like LoRA in FL can suffer from "aggregation inconsistency". Therefore, they propose UniFLoW, a unified framework with three core contributions:

1- It uses a pre-trained universal encoder (ImageBind) and a base LLM (Vicuna-7B), applying LoRA to both components.

2- Clients first fine-tune their respective encoder's LoRA parameters and then fine-tune the base LLM's LoRA parameters within a single local training round.

3- Their server-side aggregation algorithm (FedA²-LoRA) computes the global A matrix by simple averaging. Then, they find the corresponding global B matrix based on A.

The authors evaluate UniFLoW on multi-modal QA (image and audio) and the FedA²-LoRA component on unimodal NLU (GLUE).

**Strengths:**

* The Federated MLLM is an interesting problem.

* The ablation study confirms some of the choices. For example, it shows that the two stage training  is effective, yielding better results than end-to-end local fine-tuning.

* The authors show the effectiveness of their method through different experiments.

**Weaknesses:**

* The paper compares its performance against methods like FFA-LORA (which freezes $A$ and only uploads $B$) and FedSA-LORA (which only uploads $A$). These methods have half the client-to-server communication cost.

* The authors do not provide any justification for some claims for example “The A matrices capture more general information”

**Questions:**

1- Are BERTScore and Token Accuracy reliable metrics for evaluating open-domain, generative QA?

2- Line 096 is not clear. How should I read this part?

3- Figure 2 is very unclear and does not help to understand the method. The figure description is very short and does not help much.

4- It is not clear for me that if this paper is the first paper that works on first federated MLLMs fine-tuning framework (line 119) or based on the beginning of line 192 there are other FedMLLMs approaches.

---

> ### Author Response · Authors · 2025-11-21
> **Author Responses (1/2)**
>
> We sincerely thank you for the constructive and detailed feedback. We appreciate your recognition that (i) federated MLLM fine-tuning is an interesting and timely problem, (ii) the two-stage training ablation is meaningful, and (iii) the empirical results demonstrate the effectiveness of our method. Below, we address the weaknesses and questions point by point.
>
> >W1: The paper compares its performance against methods like FFA-LORA (which freezes 𝐴 and only uploads 𝐵) and FedSA-LORA (which only uploads 𝐴). These methods have half the client-to-server communication cost.
>
> We acknowledge that this method incurs twice the communication cost compared to FFA-LoRA and FedSA-LoRA. However, relative to FedEx-LoRA, which also aims to recover $\Delta_W^*$, it reduces the communication volume by a factor of $d/(2r)$, achieving a more efficient recovery scheme with lower communication overhead while maintaining accuracy. A detailed comparison of training-time communication and computational complexity is provided in the table below. We have also added this part to the **Appendix C.1**.
>
> | Method | # Trainable Params | # Per-round Communicated Params | Computational Complexity|
> |-|-|-|-|
> |LoRA |1.83M| 0.78M|O(2Krd)|
> |FFA-LoRA|1.44M| 0.39M |O(Krd) |
> |FedDPA-LoRA  | 2.62M  | 0.78M | O(Krd) |
> |FedSA-LoRA| 1.83M | 0.39M | O(Krd) |
> |FedEx-LoRA|1.83M | 25.74M | O(Krd²) |
> |$FedA²-LoRA$|1.83M | 0.78M|O(Krd²)|
>
> Where K represents the number of clients, the weight dimension of W is dxd, and the dimensions of B and A are dxr and rxd, respectively. Here, $d$ is typically {1024, 4096}, and $r$ is typically {4, 8, 16}. Therefore, $O(r^3)$ is not the dominant complexity term.
>
> >W2: The authors do not provide any justification for some claims for example “The A matrices capture more general information”
>
> Thank you for pointing this out. We have already provided a more detailed analysis of this part in **Appendix A** of the revised version. First, we theoretically derive the optimal solutions for  $A ^ *$ and $B ^ *$ as follows:
>
> $A^*= \mathbb{B}^{\dagger}\Delta_W$
>
> $B^*= \Delta_W\mathbb{E}[X_{n_k}X_{n_k}^T]\mathbb{A}^T (\mathbb{A}\mathbb{E}[X_{n_k}X_{n_k}^T]\mathbb{A}^T)^{-1}$
>
> Where $\mathbb{A}$ and $\mathbb{B}$ are Gaussian random initialization matrices.Therefore, we can derive the following relationship between A* and B*:
>
> $$
>  B^* = \mathbb{B} A^*{\mathbb{E}} [X_{n_k} X_{n_k}^T] (\mathbb{A}^{T} \mathbb{E} [X_{n_k} X_{n_k}^T] \mathbb{A}^{T} )^{-1}
> $$
>
> From the above formula, we can obtain $A ^ *$ only contains information unrelated to the data, which we consider to be general information. $B ^ *$ can be written as a function of $A ^ *$ and $X _ {n _ k}$, meaning it can capture both general and data information. Since A and B usually have the same rank, $A ^ *$ contains more general information when acquiring the same amount of information.
>
> Meanwhile, since $B ^ *$ can be written as a function of $A ^ *$ and $X _ {n _ k}$, it indicates that $ B$ also contains general information, making the aggregation of $ B$ necessary. Therefore, we propose $FedA ^ 2$-$LoRA$ to better capture global information and recovering $\Delta _ W ^ *=\frac{1}{K}\sum _ {k=1}^K B _ k  A _ k$.
>
> We also conducted experiments to analyze the similarity and sorted them from highest to lowest.  In **Appendix A**, our **Figure 4** shows some of the results from the value layer.
>
> | Layer Name | LoRA Parameter Name | Cosine Similarity |
> |:---|:---|:---|
> |15 | value.lora\_A | 0.212694 |
> |20 | value.lora\_A | 0.137847 |
> |12 | value.lora\_A | 0.117822 |
> |13 | value.lora\_A | 0.094493 |
> | 22 | value.lora\_A | 0.085506 |
> |18 | value.lora\_A | 0.083193 |
> |14 | value.lora\_A | 0.073243 |
> |19 | value.lora\_A | 0.062988 |
> |3 | value.lora\_A | 0.061706 |
> |21 | value.lora\_A | 0.048951 |
> |7 | value.lora\_A | 0.04726 |
> |1 | value.lora\_A | 0.038443 |
> |2 | value.lora\_A | 0.038439 |
> |16 | value.lora\_A | 0.035118 |
> |9 | value.lora\_A | 0.030625 |
> |4 | value.lora\_A | 0.029767 |
> |10 | value.lora\_A | 0.025706 |
> |10 | value.lora\_B | 0.025444 |
> |17 | value.lora\_B | 0.022276 |
> |23 | value.lora\_A | 0.019876 |
> |5 | value.lora\_A | 0.016306 |
> |14 | value.lora\_B | 0.015442 |
> |3 | value.lora\_B | 0.014453 |
> |11 | value.lora\_A | 0.01341 |
> |20 | value.lora\_B | 0.013018 |
> |0 | value.lora\_B | 0.010565 |
> |0 | value.lora\_A | 0.00969 |
> |6 | value.lora\_B | 0.007943 |
> |8 | value.lora\_A | 0.00707 |
> |1| value.lora\_B | 0.006977 |
> |22 | value.lora\_B | 0.006539 |
> |23 | value.lora\_B | 0.004681 |
> |11 | value.lora\_B | 0.004258 |
> |4 | value.lora\_B | 0.001911 |
> |18 | value.lora\_B | 0.001812 |
> |8 | value.lora\_B | 0.000097 |
> |5 | value.lora\_B | -0.001479 |
> |7 | value.lora\_B | -0.001675 |
> |19 | value.lora\_B | -0.002555 |
> |16 | value.lora\_B | -0.002883 |
> |21 | value.lora\_B | -0.003744 |
> |13 | value.lora\_B | -0.003967 |
> |17 | value.lora\_A | -0.005535 |
> |2 | value.lora\_B | -0.008209 |
> |15| value.lora\_B | -0.009123 |
> |9 | value.lora\_A | -0.011076 |
> |12 | value.lora\_B | -0.012479 |

---

> > ### Author Response · Authors · 2025-11-21
> > **Author Responses (2/2)**
> >
> > >Q1: Are BERTScore and Token Accuracy reliable metrics for evaluating open-domain, generative QA?
> >
> > Thank you for your question. Due to the nature of our task-based open dataset, it is difficult to measure semantic quality using traditional Exact Match/BLEU. Therefore, using both BERTScore and token accuracy can better measure model performance. On the one hand, BERTScore uses BERT to semantically encode the prediction and reference answer, and measures semantic consistency by calculating the similarity of token-level embeddings. It has been widely used in many generation tasks [1][2] and has been proven to be highly relevant to human evaluation. On the other hand, token accuracy can measure more finely whether the model can generate correct information token by token, focusing on consistency between form and content. Therefore, we evaluate from both textual and semantic perspectives.
> >
> > You may feel that our evaluation is not comprehensive enough. Here are some other metrics that were trained using both image and speech data, as shown below:
> >
> > | Method| Image BLEU | Image ROUGE‑L | Image METEOR | Audio BLEU | Audio ROUGE‑L | Audio METEOR |
> > |-|-|-|-|--|-|-|
> > | — | 31.62| 50.14 | 50.42 | 29.94 | 53.33 | 45.39 |
> > | LoRA| 32.10 | 65.00 | 57.14| 33.06| 62.50| 51.34|
> > | LoRA (II stage)| 35.45| 64.29 | **57.74** | **36.86**| 64.29| 53.29|
> > | FFA‑LoRA| 33.49 | 62.50 | 50.76| 32.06 | 65.00 | 53.57 |
> > | FFA‑LoRA (II stage)| 35.88| 65.38 | 52.24 | 35.53 | 62.50 | 55.38|
> > | FedSA‑LoRA| 31.13| 62.50 | 50.20| 31.58 | 61.90 | 50.26|
> > | FedSA‑LoRA (II stage)| 35.45| 63.33 | 51.88| 35.00| 64.29| 52.24|
> > | FedA²‑LoRA | 36.85| 64.29 | 54.59| 33.86| 65.00| 54.17|
> > | FedA²‑LoRA (II stage)| **38.96**| **66.67** | 55.21| 35.87| **67.86**| **55.88** |
> >
> > >Q2: Line 096 is not clear. How should I read this part?
> >
> > We appreciate you pointing out the ambiguity in Line 096.We confirm that the expression was unclear, and we have revised it in the latest manuscript as follows:
> > “To address the challenges of ① **unlocking private multimodal data at scale**, ② **modality‑induced architectural incompatibility**, ③ **communication overhead**, ④ **aggregation inconsistency**, and ⑤ **accuracy–efficiency trade‑off** in exploiting private multimodal client data, …”
> > The main purpose here is to explain what difficulties we need to solve in sequence.Should you have any further questions or suggestions, we would be happy to address them.
> >
> > >Q3: Figure 2 is very unclear and does not help to understand the method. The figure description is very short and does not help much.
> >
> > Thank you for your suggestion. We have redrawn Figure 2 in the latest version and included the previous versions in the appendix F for readers who need more details. In the new version, the caption has also been changed as follows:
> > "The inference process operates on each client, where local multi-modal data (e.g., text, images, or audio) is processed by modality-specific encoders (e.g., ImageBind), and the resulting features are then passed to a pretrained LLM (e.g., Vicuna-7B). The fine-tuning procedure is organized into two stages: in the I  stage, we apply LoRA to the modality encoders, and in the II  stage, we use LoRA to fine-tune the pretrained large language model. The server aggregates parameter updates from clients using the proposed $FedA^2$-$LoRA$ method, which ensures consistent and efficient updates while minimizing communication overhead. This framework is designed to handle heterogeneous data sources across clients, thereby enhancing scalability and model performance across different modalities.
> > "
> > >Q4: It is not clear for me that if this paper is the first paper that works on first federated MLLMs fine-tuning framework (line 119) or based on the beginning of line 192 there are other FedMLLMs approaches.
> >
> > Thank you for your careful reading. To be more rigorous, we have revised the statement as follows: "To fully leverage distributed multi-modal private data, we design a federated fine-tuning framework **UniFLoW** for MLLMs with universal modality support. To the best of our knowledge, this is the first federated **Universal** MLLMs fine-tuning framework that enables general-purpose modality integration." There is existing work on FedLLM fine-tuning [3][4], but they mainly focus on single-task multimodal approaches. However, we have implemented a Universal FedLLM fine-tuning using existing LLMs and encoders, which allows for more comprehensive unlocking of private data.
> >
> > If you have any further questions, we would be happy to answer them.😁
> >
> > [1]Hanna M, Bojar O. A fine-grained analysis of BERTScore. 2021.
> >
> > [2]Ao J, Wang Y, Tian X, et al. Sd-eval: A benchmark dataset for spoken dialogue understanding beyond words.2024.
> >
> > [3]Zhang J, Yang H, Li A, et al. MLLM-LLaVA-FL: Multimodal Large Language Model Assisted Federated Learning, 2025.
> >
> > [4]Zhang Y, Gao H, Chen H, et al. FedNano: Toward Lightweight Federated Tuning for Pretrained Multimodal Large Language Models, 2025.

---

> ### Author Response · Authors · 2025-11-25
> **Looking forward to your feedback**
>
> Dear Reviewer oyVC,
>
> We are eager to ensure that we have adequately addressed your concerns and are prepared to offer further clarifications or address any additional questions you may have.
>
> Should you find that our revisions have satisfactorily addressed your main concerns, we would be most grateful if you would **reconsider the evaluation of our paper to enhance its standing**.
>
> We would like to express our heartfelt gratitude for the time and effort you have dedicated to reviewing our work. It has been a pleasure to engage with you throughout this process.😁
>
> Best regards,
>
> The authors

---

### Author Response · Authors · 2025-11-23
**General Response to All Reviewers**

Dear Reviewers,

We sincerely thank all the reviewers (`oyVC`, `TTmT`, `ZbHw`, `2BaQ`) for their valuable feedback. We are encouraged that they found the federated MLLM setting interesting and practically meaningful and acknowledged the effectiveness of our Ⅱ stage training strategy and extensive experiments in validating our design choices (`oyVC`, `TTmT`). We are also glad that the reviewers appreciated the novelty and efficiency of **FedA²-LoRA** in analytically addressing LoRA aggregation inconsistency (`TTmT`, `2BaQ`), recognized the clear and sensible design of **UniFLoW** for handling heterogeneous modality data via a general-purpose encoder and LoRA-equipped key modules (`TTmT`, `ZbHw`), and valued the well-written and easy-to-understand presentation of our manuscript (`2BaQ`).

We have made every effort to respond to your comments thoroughly and accurately. Following the reviewers’ suggestions, we have incorporated the following additions and revisions:
+ **Analysis of the “General information and personalized information” assumption in $FedA^2-LoRA$**:
We added a theoretical proof of this hypothesis, along with analytical experiments under real non-IID conditions in **Appendix A** to addresses the concerns of Reviewers `oyVC` and `TTmT`.
+ **Proof of optimality for explicit ridge-regression formulation**: We added optimization objectives **Eq. 10 and Eq. 12** to **section 3.2**, and in **Appendix B**, we added how to derive our solution from the optimization to address reviewer `TTmT` and `ZbHw`'s question.
+ **Communication cost,training cost and time complexity analysis of the closed-form solution**: Regarding the communication cost issues raised by all reviewers(`oyVC`, `TTmT`, `ZbHw`, `2BaQ`), and the computational complexity analysis mentioned by reviewer `2BaQ`, we have presented the findings in Table 6 of **Appendix C.1**.
+ **Payload comparison under matched budgets**: Regarding the head-to-head comparison requested by reviewer `TTmT`, previous comparisons were based on the same computational cost. We have now made a comparison in Table 8 of **Appendix C.3**.
+ **Discussion on heterogeneity-aware variants**: In response to Reviewer `TTmT`, we attempted a zero-padding approach to implement heterogeneity-aware variants in **Appendix C.4** ,Table 9.
+ **Additional ablation studies on Tikhonov Regularization**: We conducted an ablation study addressing this concern from Reviewer `ZbHw` in Table 11 of **Appendix C.6**. In addition, **Figure 3(a)**  in Section 4.3 presents an ablation analysis of the hyperparameter $\lambda$.
+ **Alternatives to ImageBind and Vicuna**:We replaced vicuna and imagebind to resolve reviewer `2BaQ`'s doubts, and this is shown in Table 12 of **Appendix C.7**.
+ **Extended analysis of the two-stage training strategy**:Regarding the exchange of training order and the two-stage training over communication rounds proposed by Reviewer `2BaQ`, we conducted corresponding experimental analyses, which are presented in Table 11 of **Appendix C.8** and Table 12 of **Appendix C.9**, respectively.
+ **Improved chart design, symbol consistency, and writing quality:** We redesigned **Figure 2**, simplified the illustrations, and thoroughly checked the consistency of symbols throughout the paper to improve overall clarity.

We have incorporated the suggested modifications in the revised version, which are highlighted in blue. If you are satisfied with the revisions, we kindly request your approval to consider an improved score.

Thanks for all the reviewers' time again.

Best regards,

Authors

---

### Author Response · Authors · 2025-12-01
**Summary of the Discussion(1/2)**

Dear ICLR 2026 AC, SAC, PC and Reviewers,
We would like to sincerely thank you and all the reviewers (`oyVC`, `TTmT`, `ZbHw`, `2BaQ`) for the time and effort invested in evaluating our submission **UniFLoW: Universal Multi-Modal Federated LoRA Fine-Tuning Framework with Analytical Aggregation**.Unfortunately, because the discussion phase was terminated and rolled back to the pre-discussion state, we were not able to see the reviewers’ final responses to our rebuttal and proposed revisions. We would be very grateful if you could kindly check whether our current manuscript and rebuttal adequately address the main concerns raised in the reviews. To facilitate your further assessment, we have compiled a concise summary of each reviewer’s key strengths and weaknesses, together with the corresponding changes we have made and brief explanations. Thank you very much for your time and for your service to the conference.

### Strength
**Practically meaningful problem setting**:Multiple reviewers agreed that the *federated MLLM fine-tuning* scenario we study is both interesting and practically relevant for real-world deployment (`oyVC`, `TTmT`).

**Effective two-stage training strategy with solid empirical support**:Reviewers highlighted that our Ⅱ stage training strategy is well-motivated and that the extensive experiments clearly validate the key design choices and their effectiveness (`oyVC`, `TTmT`).

**Novel and efficient FedA²-LoRA for resolving aggregation inconsistency**:Reviewers appreciated the novelty and efficiency of **FedA²-LoRA**, explicitly noting that it provides an analytical solution to the long-standing issue of LoRA aggregation inconsistency in federated settings (`TTmT`, `2BaQ`).

**Clear and general UniFLoW framework for heterogeneous multimodal data**:Reviewers recognized that **UniFLoW** offers a clear and sensible design for handling heterogeneous modality data, by combining a general-purpose encoder with LoRA-equipped key modules in a unified framework (`TTmT`, `ZbHw`).

**High-quality, accessible presentation**:At least one reviewer explicitly commended the manuscript as well written and easy to follow, which they viewed as enhancing the accessibility and impact of the technical contributions (`2BaQ`).


### Weaknesses and Revisions
+ **Analysis of the “General information and personalized information” assumption in $FedA^2-LoRA$**:
We added a theoretical proof of this hypothesis, along with analytical experiments under real non-IID conditions in **Appendix A** to addresses the concerns of Reviewers `oyVC` **W2** and `TTmT` **W1**.
+ **Proof of optimality for explicit ridge-regression formulation**: We added optimization objectives **Eq. 10 and Eq. 12** to **section 3.2**, and in **Appendix B**, we added how to derive our solution from the optimization to address reviewer `TTmT`'s **W2** and `ZbHw`'s **W2**.
+ **Communication cost,training cost and time complexity analysis of the closed-form solution**: Regarding the communication cost issues raised by all reviewers(`oyVC`**W1**, `TTmT`**W3**, `ZbHw`**W1**), and the computational complexity analysis mentioned by reviewer `2BaQ` **Q3**, we have presented the findings in Table 6 of **Appendix C.1**.
+ **Payload comparison under matched budgets**: Regarding the head-to-head comparison requested by reviewer `TTmT` **W4**, previous comparisons were based on the same computational cost. We have now made a comparison in Table 8 of **Appendix C.3**.
+ **Discussion on heterogeneity-aware variants**: In response to Reviewer `TTmT` **W5**, we attempted a zero-padding approach to implement heterogeneity-aware variants in **Appendix C.4** ,Table 9.
+ **Additional ablation studies on Tikhonov Regularization**: We conducted an ablation study addressing this concern from Reviewer `ZbHw` **W4** in Table 11 of **Appendix C.6**. In addition, **Figure 3(a)**  in Section 4.3 presents an ablation analysis of the hyperparameter $\lambda$.
+ **Alternatives to ImageBind and Vicuna**:We replaced vicuna and imagebind to resolve reviewer `2BaQ` **W2**'s doubts, and this is shown in Table 12 of **Appendix C.7**.
+ **Extended analysis of the two-stage training strategy**:Regarding the exchange of training order and the Ⅱ stage training over communication rounds proposed by Reviewer `2BaQ`**Q1** and **Q2**, we conducted corresponding experimental analyses, which are presented in Table 11 of **Appendix C.8** and Table 12 of **Appendix C.9**, respectively.
+ **Improved chart design, symbol consistency, and writing quality:** We redesigned **Figure 2**, simplified the illustrations, and thoroughly checked the consistency of symbols throughout the paper to improve overall clarity.

---

> ### Author Response · Authors · 2025-12-01
> **Summary of the Discussion(2/2)**
>
> ### Misunderstandings and Clarifications
>
> **Clarifications and Revisions in Response to Reviewer Comments**
>
> **UniFLoW’s Contribution in Handling Multimodal Data**raised by reviewer `2BaQ` **W1**: We fully acknowledge that ImageBind is a robust encoder capable of handling various modalities. However, our primary contribution is in proposing a **universal multimodal federated fine-tuning framework**—**UniFLoW**—that builds on existing modality encoders and pretrained models. This approach, which leverages federated learning for fine-tuning, has not been explored in prior work. We hope that **UniFLoW** can serve as a foundation for future multimodal federated fine-tuning paradigms.
>
> **Aggregation Inconsistency and Communication Cost in FedA²-LoRA**  raised by reviewer `ZbHw` **W1**:We appreciate this insightful comment and apologize for the confusion caused by our initial explanation. You are correct that directly solving on the server could eliminate the aggregation error. However, the challenge arises when transmitting back to the client. Without decomposing into $B$ and $ A$, subsequent updates would be impossible. Additionally, for large models like 7B, the communication required for $B$ is much larger than that for $B$ and $A$ combined. To address this, our approach decomposes $B$ and $A $, reducing the communication cost by up to 128 times compared to the **FedEx-LoRA** method, which also recovers $B$.
>
> These clarifications should provide a clearer understanding of our methodology and resolve any confusion regarding communication costs and the contributions of **UniFLoW**.
>
>
> We hope that these clarifications and the revisions we have made will improve the manuscript and bring it closer to meeting the standards of ICLR. We are also very much looking forward to further discussions and would greatly appreciate any additional feedback or suggestions for improvement. Thank you for your continued support and for facilitating this process.
>
> Thank you once again to all the Program Chairs, Senior Area Chairs, Area Chairs, and Reviewers for their valuable time and thoughtful feedback.
>
> Best regards,
>
> The authors of **948**

---

### Meta-Review · Area_Chair_Yq3N · 2026-01-06

**Summary:**

This submission proposes UniFLoW, a federated fine-tuning framework for multimodal LLMs, combining a universal encoder (e.g., ImageBind) with an LLM (e.g., Vicuna-7B) and a two-stage local training procedure. Its central technical claim is FedA²-LoRA, which aims to address LoRA aggregation inconsistency by averaging the LoRA A matrices and reconstructing B via an analytic (ridge/Tikhonov) solution. Across reviewers, the paper is viewed as tackling an interesting setting, but the main reasons it falls below the acceptance bar are: (i) the key aggregation/reconstruction rationale initially lacked convincing justification, and even with added derivations the approach remains somewhat heuristic and its benefit over simpler alternatives is not fully compelling; (ii) communication-cost claims and baseline comparisons were considered insufficiently convincing and/or not aligned with the most relevant budget-matched settings; and (iii) concerns that much of the apparent “universality” and multimodal capability may largely come from the chosen backbone (ImageBind) and standard design choices, making the incremental contribution less clear.

**Reviewer Concerns:**

Concerns addressed in the rebuttal / revision:

+Formalization and missing objective/derivation: Reviewers asked for an explicit ridge objective and proof/derivation; the rebuttal indicates they added an explicit objective and derivation/proof content.

+Communication/time complexity reporting: Added tables/analysis for communication and complexity, and clarified what is communicated and why.

+Matched-budget comparison request: At least partial response by adding comparisons under matched communication cost/round budgets (including discussion around FedEx-LoRA / budget-matching).

+Ablations on Tikhonov regularization and presentation clarity: Added a regularization ablation and reported sensitivity; also claims of improved figures/writing.

Concerns that remain material (in my view):

-Core methodological necessity/convincingness: A key critique was that reconstructing B in this way is not clearly more reasonable than directly aggregating BA (or otherwise avoiding the proposed reconstruction), and that uploading both A and B does not automatically imply reduced communication in the most relevant comparisons. The rebuttal clarifies their intended comparison point (vs. FedEx-LoRA), but does not fully resolve the broader concern that the claimed advantage depends strongly on which baselines and budgets are deemed primary.

-Incremental contribution vs. backbone/design choices: A reviewer explicitly notes the paper may overstate its novelty because the “universal” multimodal aspect largely follows from ImageBind, and the remaining novelty is less sharply isolated. The rebuttal argues “first universal FedMLLM fine-tuning framework,” but the incremental technical leap still feels borderline for ICLR without stronger, cleaner evidence that UniFLoW (beyond backbone choice + two-stage training) is necessary and general.

-Evaluation strength and clarity: Concerns remain around whether the evaluation (metrics like BERTScore/token accuracy for open-ended QA) and the breadth of experimental evidence are sufficient to support the paper’s strongest claims of universality and practicality.

**Reviewer Scores:**

Given that current discussion and rebuttal, my best estimate is that rebuttal-driven updates might slightly improve sentiment, but not enough to reliably cross the threshold.

---

### Decision · Program_Chairs · 2026-01-26

Reject